# Hybrid Sol–Gel Matrices Doped with Colorimetric/Fluorimetric Imidazole Derivatives [note 1]

**DOI:** 10.3390/nano11123401

**Published:** 2021-12-15

**Authors:** Rui P. C. L. Sousa, Rita B. Figueira, Bárbara R. Gomes, Sara Sousa, R. Cristina M. Ferreira, Susana P. G. Costa, M. Manuela M. Raposo

**Affiliations:** 1Centre of Chemistry, Campus of Gualtar, University of Minho, 4710-057 Braga, Portugal; rui.sousa@quimica.uminho.pt (R.P.C.L.S.); barbara.sgomes11@gmail.com (B.R.G.); sarah.sousa00@hotmail.com (S.S.); crisferreira44103@hotmail.com (R.C.M.F.); spc@quimica.uminho.pt (S.P.G.C.); mfox@quimica.uminho.pt (M.M.M.R.); 2Vasco da Gama CoLAB, Rua Roberto Frias, 4200-465 Porto, Portugal

**Keywords:** organic–inorganic hybrid, sol–gel, imidazole derivatives, fluorescence, colorimetry

## Abstract

Organic–inorganic hybrids (OIH) are materials that can be easily synthesized by the sol–gel method and combine the advantages of organic and inorganic moieties within a single polymeric matrix. Imidazole derivatives are versatile organic compounds that can change their optical properties with the variation of pH due to the protonation or deprotonation of the nitrogen atoms. This work reports the preparation of different OIHs doped with different contents of two imidazole compounds (**3a**,**b**). The obtained materials were characterized structurally by FTIR, and the dielectric properties were studied by electrochemical impedance spectroscopy. The optical properties were studied by UV-Vis absorption and fluorescence spectroscopies. The FTIR analysis showed that the presence of the imidazole does not change the structural properties of the matrices. The normalized resistance values obtained for the doped matrices ranged between 8.57 and 9.32 Ω cm^2^, all being higher than the undoped matrix. The σ ranged between 9.49 and 10.28 S cm^−1^, being all higher than the pure OIH samples. Compound **3a** showed a maximum absorption peak at 390 nm, which is present in the OIH spectra, proving the presence of the compound. In the case of compound **3b**, a maximum absorption wavelength at 412 nm was found, and the compound peak was not clear, which may indicate that an interaction between the compound and the matrix occurred. A synergetic effect between the intrinsic emission of the matrix and the fluorescence of **3a** is found on the OIH-doped matrices.

## 1. Introduction

The sol–gel method is a synthetic procedure that allows obtaining multifunctional materials in different forms e.g., monoliths, aerogels, xerogels, and films, among others. This method has been used since the early 20th century [1], although only in the last few decades has this procedure been widely explored and implemented. The versatility of the sol–gel method is directly linked to the numerous variables that can influence the final form or properties of a material, e.g., the type of precursor used, the substrate or mold in which the sol is stored, the solvent used, the pH, the temperature, and reaction time, as well as the type of drying treatment [2]. Organic–inorganic hybrids (OIH), also known as ORMOSILs, are one common type of final materials that can be easily obtained by this method joining the synergies of organic and inorganic moieties within the same polymeric network [3].

Polyetheramines are a type of organic moieties generally used in the synthesis of Class II OIHs, since the amine terminal group of this chain can be linked through a covalent bond with silanes functionalized with isocyanate or epoxy groups. Commercial polyetheramines, namely Jeffamines, have widely been used in the preparation of several OIH materials [4,5,6,7,8,9,10,11]. The presence of Jeffamines within sol–gel matrices provide the hydrophobicity and flexibility of carbon chains, yielding to the formation of ureasilicates or aminoalcoholsilicates, according to the bond established between the terminal amine group and the functional group of the silane used. Moreover, one of the main advantages of OIHs is that this type of material can be easily doped with chemosensors, molecules sensitive to a certain analyte, allowing obtaining sensitive membranes.

Imidazole derivatives are a class of organic compounds known for their versatility. The imidazole ring is amphoteric, which confers interesting properties to imidazole derivatives. The imidazole ring can be protonated or deprotonated, with subsequent changes in the optical properties (see Figure 1).

Depending on the functionalization of the imidazole ring, the -NH proton acidity can be changed. Together with the capacity of selectively binding cationic species, this turns imidazoles into potential optical chemosensors for a wide range of applications [12] such as pH [13,14] and cation detection [15,16]. In addition to this application, imidazole derivatives have also found application as thermally stable fluorophores for Organic Light-Emitting Diode (OLED)s [17], two-photon absorbing molecules [18], nonlinear optical materials [19], DNA intercalators [20], and PDI photosensitizers [21]. Moreover, the introduction of π bridges of a heterocyclic nature, such as sulfur five-membered heterocycles, can improve the photophysical properties and provide additional binding sites for certain ions.

In the last few years, several imidazole derivatives have been doped in sol–gel materials. For instance, Zhang et al. produced electroactive poly(ionic liquid) gels through the sol–gel method, by copolymerization of an imidazolium-type ionic liquid with 2-(dimethyl amino)ethyl methacrylate and poly(ethyleneglycol) diacrylate as the cross-linker [22]. The reported material is responsive to CO_2_, with reversible protonation/deprotonation of the imidazole ring, resulting in a reversible sol to gel transition. Qian et al. prepared a cross-linked siloxane with phosphoric acid, 3-glycidoxipropyltrimethoxysilane (3-GPTMS) and polybenzimidazole, to act as a proton exchange membrane [23]. Tarley et al. developed an OIH based on 1-vinylimidazole, (3-aminopropyl)trimethoxysilane (APTMS), vinyltrimethoxysilane, tetraethoxysilane (TEOS), and 2,2′-azoisobutyronitrile [24]. The produced material showed high As(V) adsorption. Marins and Soares also established the achievement of an OIH doped with an imidazole moiety, based on 1-(3-trimethoxysilylpropyl)-3-methylimidazolium chloride (IL-TMOS), for application as an electrorheological fluid [25]. The aforementioned work showed the advantage of low leaching of the imidazole when compared to other similar studies. Another OIH with an imidazole moiety was produced by Khristenko et al. [26]. In this work, 1-*n*-propyl-3-methylimidazolium chloride was fixed on a silica matrix, providing high and heterogeneous polarity and acidity to the surface of the material. Sabeela et al. produced a silica imidazolium ionic liquid, based on (3-aminopropyl)triethoxysilane (APTES). The structure showed high chemical adsorption capacities for organic water pollutants, namely Coomassie blue (CB-R250) [27]. 1,2-Dimethylimidazole was used in the preparation of another OIH material, together with poly(2,6-dimethyl-1,4-phenylene oxide, 3-GPTMS, and TEOS [28]. The matrix was developed to be applied on alkaline fuel cells, and the used precursors improved the conductivity and alkaline resistance.

Hosseini and Mohebbi, in 2020, produced TiO_2_ nanoparticles modified with histidine and imidazole by the sol–gel method [29]. The modifiers were added to a solution of TiO_2_ nanoparticles dispersed in a dilute sulfuric acid solution. The produced materials were able to perform a photocatalytic reduction of aqueous Zn^2+^, Pb^2+^, and Cu^2+^ ions. Ribicki et al. developed an OIH functionalized with an imidazole group [30]. The material was based on the reaction between 3-chloropropyltrimethoxysilane, an imidazole ring, and TEOS. The obtained OIH was able to adsorb Cu^2+^, Cd^2+^, and Ni^2+^ ions with high affinity. The literature reported so far with imidazole moieties in sol–gel OIH materials showed that this type of material is suitable for several applications, such as fluids for fuel cells, cation/pollutants adsorption, or optical fiber sensors [31], and it has high affinity to different molecules, proving its versatility and multifunctionality.

In this work, the authors report the synthesis and characterization of two imidazole derivatives (**3a**,**b**), following the group’s works for different applications [18,32,33]. The synthesis and characterization of the imidazole derivative **3b** is reported for the very first time. New OIH sol–gel materials doped with different contents of **3a**,**b** compounds are reported. The OIH sol–gel matrix used was based on Jeffamine^®^ THF-170 and 3-GPTMS [34]. The influence of the contents of the two imidazole derivatives (**3a**,**b**) was studied on the dielectric and optical properties of the doped OIH materials. The structural properties were assessed by Fourier transform infrared spectroscopy (FTIR). The materials developed and reported in this article set the roots in the development of optical fiber sensors to monitor pH changes in the range between 9 and 12.5. This pH range allows, for instance, its application in concrete structures’ health monitoring.

## 2. Experimental Section

### 2.1. Materials

Reaction progress was monitored by thin layer chromatography (0.25 mm thick precoated silica plates: Merck Fertigplatten Kieselgel 60 F254, Merck KGaA, Darmstadt, Germany), while purification was performed by silica gel column chromatography (Merck Kieselgel 60; 230–400 mesh, Merck KGaA, Darmstadt, Germany). NMR spectra were obtained on a Varian Unity Plus Spectrometer (Varian, Palo Alto, CA, USA) at an operating frequency of 300 MHz for ^1^H and 75.4 MHz for ^13^C or a Bruker Avance III 400 (Bruker, Billerica, MA, USA) at an operating frequency of 400 MHz for ^1^H and 100.6 MHz for ^13^C using the solvent peak as internal reference. The solvents are indicated in parenthesis before the chemical shift values (*δ* relative to TMS and given in ppm). Melting points (Mps) were determined on a Gallenkamp apparatus (Gallenkamp, Cambridge, UK). Infrared spectra were recorded on a BOMEM MB 104 spectrophotometer (ABB, Zurich, Switzerland). Mass spectrometry analyses were performed at the “C.A.C.T.I.—Unidad de Espectrometria de Masas” at the University of Vigo, Vigo, Spain. UV-visible absorption spectra (200–700 nm) were obtained using a Shimadzu UV/2501PC spectrophotometer (Shimadzu, Kyoto, Japan). All commercially available reagents were used as received. The relative fluorescence quantum yield of **3b** was determined by using a 10^−6^ M solution of 9,10-diphenylanthracene (DPA) in ethanol as standard (*Φ_F_* = 0.95) [35]. For the *Φ_F_* determination, the fluorescence standard was excited at the wavelength of maximum absorption found for the compound to be tested, and in the fluorimetric measurement, the absorbance of the solution did not exceed 0.1. The synthesis of imidazole **3a** was reported elsewhere [18].

Commercial reagents Jeffamine^®^ THF-170 (Huntsman Corporation, Pamplona, Spain), 3-glycidoxipropyltrimethoxysilane (3-GPTMS) (97%, Sigma-Aldrich, St. Louis, MO, USA), Jeffamine^®^ THF-170 (Huntsman Corporation, Pamplona, Spain), calcium hydroxide (Ca(OH)_2_, 95%, Riedel, Bucharest, Romania), and potassium hydroxide (KOH, 90%, PanReac, Darmstadt, Germany), and solvents tetrahydrofuran (99.5% stabilized with ≈300 ppm of BHT, Panreac, Darmstadt, Germany) and absolute ethanol (EtOH, PanReac, Darmstadt, Germany) were used as received. Commercial buffer solutions of pH 4, 7, and 9 (PanReac, Darmstadt, Germany) were used in the preliminary tests. High-purity deionized water with high resistivity (higher than 18 MΩ cm) obtained from a Millipore water purification system (Milli-Q^®^, Merck KGaA, Darmstadt, Germany) was used.

### 2.2. Synthesis of 2-(5′-(4″-nitrophenyl)furan-2′-yl)-1H-phenanthro[9,10-d]imidazole ***3b***

The heterocyclic aldehyde 2b (1 mmol), 9,10-phenanthrenedione 1 (1 mmol), and NH_4_OAc (20 mmol) were dissolved in glacial acetic acid (5 mL), followed by stirring and heating at reflux for 8 h. Then, the mixture was cooled to room temperature, after which ethyl acetate was added (15 mL) and washed with water (3 × 10 mL). After drying with anhydrous MgSO_4_, the solution was filtered, and the solvent was evaporated to dryness. The resulting solid was dissolved in acetone and precipitation with petroleum ether afforded the pure imidazole derivative 3b as an orange solid (120 mg, 80%). Mp > 300 °C. IR (KBr): ν = 3584, 2362, 1598, 1514, 1427, 1343, 1329, 1109, 851, 799, 745, 715, 689 cm^−1^. 1H NMR (DMSO-d6): δ = 7.65–7.80 (m, 6H, H-2, H-7, H-2″, H-6″, H-3′, and 4′), 8.25 (d, J = 9.2 Hz, 2H, H-3, and H-6), 8.38 (d, J = 8.8 Hz, 2H, H-1, and H-8), 8.67 (d, J = 7.8 Hz, 2H, H-3″, and H-5″), 8.89 (d, J = 8.4 Hz, 2H, H-4, and H-5) ppm. 13C NMR (DMSO-d6): δ = 112.2, 113.2, 122.3, 123.6, 124.1, 124.5, 125.6, 125.6, 127.0, 127.8, 131.0, 135.1, 140.1, 145.8, 146.2, 151.9, and 152.6 ppm. MS (EI) *m*/*z* (%): 405 ([M]+, 8). HRMS: (EI) *m*/*z* (%) for C_25_H_15_O_3_N_3_; calcd 405.1113; found 405.1094.

### 2.3. Synthesis of Pure OIH Films and Doped with Imidazole Derivatives ***3a**,**b***

The synthesis of pure OIH films was performed following the steps described previously [34]. In short, the Jeffamine^®^ THF170 dissolved in tetrahydrofuran (THF) and 3-GPTMS were added to a glass container using a molar ratio of 1 Jeffamine^®^ THF170:2 of 3-GPTMS. The solution was stirred for 20 min, and then, water was added. The OIHs doped used the same steps described previously; however, after 10 min, imidazole derivatives **3a**,**b** (ethanolic solution of 10^−4^ M, with four different volumes (500, 600, 900, and 1000 µL)) were added to the solution, which was stirred until a homogeneous mixture was obtained. The four gels doped with different contents of imidazole derivatives **3a**,**b** were cast into Teflon^®^ molds, covered with Parafilm^®^, and placed in an oven (UNB 200, Memmert, Buechenbach, Germany). The OIH gels in the covered molds were kept at 40 °C for 15 days to ensure that all the remaining solvents are eliminated. The pure OIHs were synthesized for comparison purposes [34].

### 2.4. Preparation of Simulated Concrete Pore Solution

A simulated concrete pore solution (SCPS) with a pH of 12.5 was prepared using deionized water at room temperature. The procedures used was according to M. Sanchez et al. and F. J. Recio et al. [36,37]. The SCPS was obtained by the addition of 0.2 M KOH to a Ca(OH)_2_ saturated solution.

### 2.5. Fourier Transform Infrared Spectroscopy (FTIR) Analysis

FTIR spectra for the imidazole doped films were recorded in transmittance mode on a Bomem MB104 spectrometer by averaging 20 scans at a maximum resolution of 4 cm^−1^. Spectra were obtained in the 4000–500 cm^−1^ range on KBr pellets. KBr pellets were prepared with 1 mg of OIH films and 100 mg of KBr.

### 2.6. Electrochemical Impedance Spectroscopy (EIS) Analysis

EIS measurements were carried out at room temperature in a Faraday cage, using a potentiostat/galvanostat/ZRA (Reference 600+, Gamry Instruments, Warminster, PA, USA), to characterize the resistance, electrical conductivity, electric permittivity, and capacitance of OIH disc materials. Films were placed between two parallel Au electrodes (10 mm ø and 250 μm thickness) using a support cell as reported in previous studies [6]. The studies were accomplished by applying a 10 mV (peak-to-peak, sinusoidal) electrical potential within a frequency range from 1 × 10^6^ Hz to 0.01 Hz (10 points per decade) at open circuit potential. The frequency response data of the studied electrochemical cells were displayed in a Nyquist plot. Gamry ESA410 Data Acquisition software (v7.8, Gamry Instruments, Warminster, PA, USA) was used for data acquisition and fitting.

### 2.7. Optical Analysis

UV-Vis spectra for both pure and doped OIH films were recorded in absorbance mode on a Shimadzu UV-2501 PC spectrophotometer (Shimadzu, Kyoto, Japan). Spectra were obtained in the range of 300–700 nm for solid samples. Fluorescence spectra for both pure and doped OIH films were recorded on a Fluoromax–4 Spectrofluorometer of Horiba Jovin Yvon (Horiba, Kyoto, Japan). Spectra were obtained in the range of 300–700 nm, with different excitation wavelengths and acquired at front-face geometry at room temperature.

## 3. Results and Discussion

### 3.1. Synthesis and Characterization of Imidazole Derivatives ***3a**,**b***

Imidazole derivative **3a** was synthesized through the Radziszewski reaction of 9,10-phenanthrenedione **1** and heterocyclic aldehyde **2a**, as previously reported [18]. Compound **3b** was synthesized by the same method, by reacting 9,10-phenanthrenedione **1** with the heterocyclic aldehyde **2b** and ammonium acetate in refluxing glacial acetic acid for 8 h. The pure imidazole derivative **3b** was obtained as an orange solid with a yield of 80% after recrystallization from acetone/petroleum ether (Figure 2). The new compound was completely characterized by NMR, IR, and UV-vis spectroscopies. The absorption (λ_max_ = 390 nm) spectra of imidazole **3b** was measured in acetonitrile solution. The study of the fluorescence of compound **3b** was also performed in acetonitrile solution using DPA as standard [35]. Unlike imidazole **3a** (*Φ_F_* = 0.50) [18], **3b** does not show fluorescence. The absence of detectable fluorescence for the push–pull imidazole **3b** functionalized with the nitro group is probably due to a high rate of S_1_ → S_0_ internal conversion, which may be related with the considerable charge-transfer character of the excited state, as a result of the strong electron-withdrawing power of the –NO_2_ group [38].

The optical response of the two compounds using the SCPS (pH = 12.5) was tested in tetrahydrofuran (THF), as it was the solvent used in the synthesis of OIH materials. In alkaline environment, the imidazole ring becomes deprotonated, causing changes in the optical properties. This is due to the higher electron density at the imidazole ring and the bands related to π → π* electronic transitions, according to the calculated ε values, which are shifted to longer wavelengths. In the case of compound **3a**, a bathochromic shift in the fluorescence spectra (λ_exc_ = 390 nm) is observed, with the maximum fluorescence wavelength changing from 455 to 497 nm. For compound **3b**, a bathochromic shift is observed in the absorbance spectra, with the maximum absorbance wavelength changing from 413 to 482 nm, causing the change of color from yellow to red. This change allows a clear naked-eye detection of pH variations in this range (see Figure 1).

### 3.2. Synthesis of OIH Films Doped with Imidazole Derivatives ***3a**,**b***

Previous studies showed that OIH materials based on Jeffamine^®^ THF-170, a diamine based on a poly(tetramethylene ether glycol) (PTMEG)/polypropylene glycol (PPG) copolymer, and 3-GPTMS showed suitable dielectric properties and resistance to a highly alkaline environment [34]. Other studies proved that doping OIH with triarylimidazole did not change the structural properties of the OIH material [39]. In this work, OIH sol–gel materials based on the aforementioned precursors were doped with two different imidazole derivatives in order to confer properties that include optical properties to the material, namely pH detection. The first step of the synthesis involves the formation of an aminoalcohol sol–gel precursor, from the reaction between the terminal amine group of the polyetheramine and the epoxy ring of 3-GPTMS. The molar ratio between Jeffamine^®^ THF-170 and 3-GPTMS is 1:2, corresponding to a NH_2_/epoxy ratio of 1:1. This means that the total consumption of all terminal groups is expected at a full conversion point. Then, this precursor is hydrolyzed, with the removal of alkoxy groups from the silane. Condensation of the silanol groups formed yield a cross-linked material in the form of a gel. Ethanolic solutions of the imidazole derivatives (1 × 10^−4^ M) were added to the solution in different volumes (500, 600, 900, and 1000 µL) during these steps to afford an improved dispersion of the dopant in the matrix. Although all epoxy groups are expected to be consumed, a reaction between the NH group from the imidazole ring and the epoxy group of 3-GPTMS that has not reacted with the Jeffamine may have occurred. This would contribute to increasing the cross-linking density of the OIH sol–gel material. The synthesis steps are summarized in Figure 2. Then, the gels were casted in Teflon^®^ molds followed by the curing step in an oven for 15 days at 40 °C. The obtained OIH films were identified as A(170)_imidazole@value of the added volume i.e., A(170)_**3a**@500, A(170)_**3a**@600, A(170)_**3a**@900, A(170)_**3a**@1000, A(170)_**3b**@500, A(170)_**3b**@600, A(170)_**3b**@900, and A(170)_**3b**@1000.

### 3.3. FTIR Analysis

Fourier transform infrared spectroscopy was used to assess the structural properties of the OIH films. The OIH matrix and precursor Jeffamine THF170 are shown in Figure 3.

The eight doped OIH films were analyzed, and the spectra signals were assigned to the corresponding structural bonds, namely: A(170)_**3a** OIHs, i.e., (i) A(170)_**3a**@500; (ii) A(170)_**3a**@600; (iii) A(170)_**3a**@900; and (iv) A(170)_**3a**@1000) spectra are shown in Figure 4; and A(170)_**3b** OIHs, i.e., (i) A(170)_**3b**@500; (ii) A(170)_**3b**@600; (iii) A(170)_**3b**@900; and (iv) A(170)_**3b**@1000) spectra are shown in Figure 5.

All spectra shows bands at 2940, 2856/2858, and 2797 cm^−1^, which are characteristic of C-H stretching vibrations [40]. These signals are typical of the organic backbone from the Jeffamine moiety and are also present in the precursor THF170 and pure hybrid A(170) spectra (Figure 3), which is according to the literature [39]. Bands at 1447 and 1369–1371 cm^−1^ are also assigned to the carbon backbone of the polyetheramine (simple bending vibrations), confirming that the backbone structure has not suffered any changes [40], which is also according to previous studies reported [39]. It can be observed that the characteristic signals of the terminal groups of both precursors (3-GPTMS–epoxy ring, 3050 cm^−1^ [41]; Jeffamine THF 170–NH terminal group, 1578 cm^−1^) [39] are not present, showing that the reaction was successfully achieved. The presence of the C–NH–C bond bending vibration, formed between the two precursors at 1655 cm^−1^, also confirms this, similarly to what was observed on pure A(170) spectra (Figure 3).

For the new signals, a broad band is present at 3400–3500 cm^−1^ and can be assigned to O–H bonds from water molecules that can still be entrapped within the polymeric hybrid and O–H groups in the polyether chain or from the aminoalcohol bond formation [41]. The peak at 1245–1261 cm^−1^ can be assigned to C–Si bond symmetric bending [42], and the large peak at 1112–1114 cm^−1^ is related to the typical Si–O–Si bonds of this cross-linked OIH material [42].

In a previous work [39], it was shown that the presence of imidazole dopants does not change the OIH material structure. In this work, that feature is also shown, and it can also be observed that no significant changes are present between the different materials’ spectra or even between doped materials’ spectra and pure A(170) (Figure 3). Moreover, since the molar ratio of the dopant compared to the precursors is small, no signals from the imidazole can be identified on all spectra.

### 3.4. Electrochemical Impedance Spectroscopy (EIS) Analysis

EIS analysis was used for the assessment of the dielectric properties (conductivity, capacitance, and electric permittivity) of the synthesized OIH materials, since this technique is a valuable one in the characterization of the material’s electrochemical properties [43,44]. Figure 6 shows the Nyquist plot obtained for the pure OIH matrix A(170) [34]. The Nyquist plots for the eight OIH doped materials as well as the fitting results are shown in Figure 7 and Figure 8. The equivalent electrical circuits (EEC) used for each sample were also introduced as an inset in each plot. The Nyquist plots show that the capacitive response over a wide range of frequencies, in which the high-frequency data are assigned to the dielectric properties of the material. The dielectric properties may indicate the potential of OIH materials to be used in alkaline samples since resistances values above 10^7^ Ω cm^2^ are generally considered suitable for high pH levels such as the ones found in fresh concrete [45,46].

Figure 7 and Figure 8 show that all the A(170) doped films describe a semi-circle that intersects the *x*-axis. In some cases (Figure 7i and Figure 8ii), the spectra indicates a dissimilar electrochemical process present at lower frequencies, which may be assigned to the Au|OIH material interface [34,45].

The analysis of the data is shown in Table 1 and was based on the EEC, which is indicated as the inset in each plot. The resistance of the film (R_Sample_) and the constant phase elements (CPE) are shown. Triplicates of each sample were performed, and the results are shown as the average of the three values. Constant phase elements (CPE) were used for seven of the eight materials instead of pure capacitance, since the results obtained do not show an ideal behavior. In the case of A(170)_3b@1000, the results show an ideal behavior, and the interfacial capacitance (C_eff_) was directly extracted from the fitting data. In the other cases, C_eff_ was calculated by using Equation (1) [47]:C_eff_ = [Q × R_sample_^(1−α)^]^1/α^(1)

The resistances of the doped OIH show values ranging between 10^8^ and 10^9^ Ω, which are higher than the ones obtained for the OIH pure matrix A(170). The percentage error is below 10% in all cases. CPE values are all in the 10^−12^ magnitude order. $\chi^{2}$ shows the goodness of the fitting, and all the values are in the 10^−3^ magnitude order, which indicates that the model used in each case is suitable.

Table 2 show the values of resistance (R), capacitance (C), conductivity (σ), and relative permittivity (ε_r_) obtained for the eight materials synthesized, which were determined using Equations (2)–(5), respectively. The values were normalized to the cell geometry dimensions, with A_Au_ standing for the gold electrodes’ area, d_Sample_ for the thickness of the sample, and ε_0_ for vacuum permittivity.
R = R_sample_ × A_Au disc_(2)
C = C_eff_/A_Au disc_(3)
σ = (d_sample_/A_Au disc_)/R_sample_(4)
ε_r_ = ((C_eff_ × d_sample_)/ε_0_) × A_Au disc_(5)

Table 2 shows that the normalized resistance values obtained are all higher than the pure A(170) matrix. Previous studies of A(170) matrices doped with organic-based molecules have already shown that doped matrices showed higher resistance than the pure matrix [39]. C values are significantly higher than the pure matrix [34], and the ε_r_ and σ values show the same behavior. All the resistances obtained for the OIH-doped sol–gel materials are above 10^7^ Ω cm^2^. According to previous studies [34,45,46], resistance values above the mentioned value imply that the synthesized OIH doped sol–gel materials can endeavor the harsh conditions of the fresh concrete. On the other hand, the reaction of imidazole NH groups with residual epoxy groups, from 3-GPTMS, may be the reason to explain the increase of the electrical resistance of the doped OIH sol–gel materials, due to an increase in the cross-linking density [48]. Conversely, the formation of silsequioxane cages within the networks is another cause that may justify the increase in the electrical resistance of the produced OIH materials. The formation of silsequioxane cages within the OIH networks leads to the reduction of the number of SiOH present, which is also due to the increase of the cross-linking density of the material and therefore higher resistance values [49]. Considering the significant difference between the values obtained for the pure OIHs and the doped ones (see Table 2), the results indicate that a synergetic behavior occurred. However, no further conclusions can be drawn. Moreover, the results here reported are in agreement with the ones obtained previously for other imidazole-doped materials [39,50].

### 3.5. Optical Analysis

Previous studies showed the potential of A(170) OIH matrices to be doped with chemosensors due to the transparency of the samples in the visible region [39]. This type of matrix was already doped with a triarylimidazole with no changes in its intrinsic properties. In this work, colorimetric and fluorimetric pH imidazole chemosensors were used to dope the A(170) films, to assess whether the doping changes the properties of the matrix or not, which was analyzed by UV/Vis spectroscopy. The OIH’s spectra shown in Figure 9 were normalized to the thickness of each film and the UV/Vis spectra of pure A(170) sol–gel film is shown in the inset. The spectra of A(170)_**3a**@1000 and A(170)_**3b**@1000 are shown in Figure 9.

Figure 9 shows that both spectra have high absorbance in the UV region. Compound **3a** show a maximum absorption peak at 390 nm, which is present in the OIH spectra, proving the presence of the imidazole compound. In the case of compound **3b**, which shows a maximum absorption wavelength at 412 nm, the compound peak is not clear, which may indicate that an interaction between the compound and the matrix occurred. Compound **3a** shows fluorimetric changes with pH variation, so A(170)_**3a** matrices were studied by fluorescence spectroscopy. Figure 10 shows the Photoluminescence spectra obtained for pure OIH material.

Figure 11 shows the photoluminescence spectra obtained for the four A(170)_**3a** OIH matrices. The spectra of A(170)_**3a**@500 (Figure 11i) show a fluorescence profile that is similar with the different excitation wavelengths. The first band seen at around 440 nm may belong to the intrinsic emission of the matrix, but **3a** also contributes to increase the fluorescence of the OIH materials. The peak of **3a** is not well defined, which is expected, since OIH materials are a biphasic system i.e., organic, and inorganic components are mixed at nanometric scale, and the presence of imidazole may contribute to a higher light scattering that may lead to changes within the sample and to a general broadening of the peaks. In the other spectra, there can be seen a higher proportion between the main peak at around 460 nm and the peak corresponding to the matrix. In all the cases, a synergetic effect between the intrinsic emission of the matrix and the fluorescence of **3a** is seen on the fluorescence profile of the doped matrices.

## 4. Conclusions

The results allow concluding that the two imidazoles reported, **3a** and **3b**, can be successfully doped in the sol–gel Jeffamine-based OIH matrix. The synthesis of 2-(5-(4-nitrophenyl)furan-2-yl)-1H-phenanthro[9,10-d]imidazole **3b** is reported for the first time, through the Radziszewski reaction, with a yield of 80%. Through the FTIR analysis, it can be concluded that doping the OIH network with the synthesized imidazoles does not change the structural properties of the matrices. Moreover, considering the electrochemical results, particularly the resistance values obtained (i.e., above 10^7^ Ω cm^2^), it can be concluded that the doped matrices are suitable to be employed in highly alkaline environments such as concrete. The optical properties of the produced OIH membranes are suitable to be used in optical fiber sensors. Nevertheless, further studies must be conducted to assess and validate the performance of these new OIH materials in contact with different pH environments, particularly in the range between 9 and 12.5. Overall, the results reported herein set the roots for the development of optical fiber sensors functionalized with OIH sol–gel membranes doped with imidazole derivatives for pH monitoring in concrete structures.

## Data Availability

Data can be available upon request from the authors.

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
