# Peer review of "Hybrid Sol–Gel Matrices Doped with Colorimetric/Fluorimetric Imidazole Derivatives [Author-notes fn1-nanomaterials-11-03401]"

_nanomaterials, 2021, doi:10.3390/nano11123401_

Round 1
Reviewer 1 Report
Submitted manuscript is describing development and characterisation of hybrid sol-gel matrices doped with pH sensitive imidazole derivatives. The aim is to produce sensor materials for optical assessment of pH changes in concrete structures. Authors have thoroughly characterised materials using FTIR and impedance spectroscopy as well as UV-Vis absorption and fluorescence spectroscopy.
However, the functional sensing characterisation of doped materials has not been performed and there is no evidence that materials could serve its intended purpose for pH sensing. Therefore, the manuscript in its present form is not complete and does not provide new insights into functional behaviour of doped hybrid materials.
Functional pH sensing characterisation of the materials should be performed and compared with the pH response of imidazole molecules in solution (calibration plots, response time, reversibility etc.). Only then the authors could discuss real potential of materials for pH sensing.
The following statement “ The OIH materials showed potential to be applied for pH detection in the range of 9 and 12.5 which are attractive properties to be used as sensor membranes in concrete structures, among other application.” is completely unsupported by results shown in this study and is very misleading.
The manuscript is clearly structured and the results are well presented following a very similar pattern used in many previously published papers from the same group. The level of novelty in this paper in comparison with their previously published works is insignificant (ref. 41, 42, 43, 47) and still does not demonstrate the optical pH sensitivity of the hybrid material with a single experiment. However, the pH sensitivity of similar materials has not been shown in the studies reported in following references either: ref 41 - PDMS Based Hybrid Sol-Gel Materials for Sensing Applications in Alkaline Environments: Synthesis and Characterization; ref 47 – Organic Inorganic Hybrid Sol-Gel Materials Doped with a Fluorescent Triarylimidazole Derivative. I find this also very misleading.
Self-citation is excessive.
Authors are encouraged to study and compare their characterisation methodology, especially related to pH sensing with some highly cited WoS papers identified with following keywords – “hybrid sol-gel” and “pH indicators”. They should not limit their search to the last 5 years only - there are many excellent review papers and original studies in the field which were missed, more recent as well as older ones.
Critical analysis of results is missing – comparison with similar functional materials? There are many papers on optical pH sol-gel sensors for concrete published but authors are not citing them.
I don’t recommend the publication of this paper unless the authors present some results demonstrating optical pH sensitivity of the materials. These should be relatively quick and straightforward experiments with great benefits for the manuscript.
Author Response
The authors would like to acknowledge all the reviewers involved for their time and dedication in reviewing the article entitled “Hybrid sol-gel matrices doped with colorimetric/fluorimetric imidazole derivatives”. All the comments are very valuable and were considered carefully in order to improve the quality and readability of the manuscript.
Reviewer 1 Comments:
Submitted manuscript is describing development and characterisation of hybrid sol-gel matrices doped with pH sensitive imidazole derivatives. The aim is to produce sensor materials for optical assessment of pH changes in concrete structures. Authors have thoroughly characterised materials using FTIR and impedance spectroscopy as well as UV-Vis absorption and fluorescence spectroscopy.
However, the functional sensing characterisation of doped materials has not been performed and there is no evidence that materials could serve its intended purpose for pH sensing. Therefore, the manuscript in its present form is not complete and does not provide new insights into functional behaviour of doped hybrid materials.
Functional pH sensing characterisation of the materials should be performed and compared with the pH response of imidazole molecules in solution (calibration plots, response time, reversibility etc.). Only then the authors could discuss real potential of materials for pH sensing.
Author’s answer: The authors agree with the reviewer. Therefore, the manuscript was rewritten focus only on the synthesis and characterization of these new materials. All the changes are highlighted yellow through the manuscript.
The following statement “ The OIH materials showed potential to be applied for pH detection in the range of 9 and 12.5 which are attractive properties to be used as sensor membranes in concrete structures, among other application.” is completely unsupported by results shown in this study and is very misleading.
Author’s answer: The authors agree with the reviewer and the statement above was removed through the manuscript.
The manuscript is clearly structured and the results are well presented following a very similar pattern used in many previously published papers from the same group. The level of novelty in this paper in comparison with their previously published works is insignificant (ref. 41, 42, 43, 47) and still does not demonstrate the optical pH sensitivity of the hybrid material with a single experiment. However, the pH sensitivity of similar materials has not been shown in the studies reported in following references either: ref 41 - PDMS Based Hybrid Sol-Gel Materials for Sensing Applications in Alkaline Environments: Synthesis and Characterization; ref 47 – Organic Inorganic Hybrid Sol-Gel Materials Doped with a Fluorescent Triarylimidazole Derivative. I find this also very misleading.
Self-citation is excessive.
Author’s answer: The authors reduced the number of self-citation. Moreover, it has been proved previously that OIH sol-gel materials with resistances values above 107 cm2 are suitable and resistant to be used in the harsh conditions of the fresh concrete. This information was introduced within the manuscript to clarify the readers and avoid misleading interpretations. All information is highlighted in yellow.
Authors are encouraged to study and compare their characterisation methodology, especially related to pH sensing with some highly cited WoS papers identified with following keywords – “hybrid sol-gel” and “pH indicators”. They should not limit their search to the last 5 years only - there are many excellent review papers and original studies in the field which were missed, more recent as well as older ones.
Author’s answer: The authors appreciate the reviewer comments. However, the manuscript now submitted is focused on OIH sol-gel materials doped with imidazole compounds. The authors are aware that there are excellent review papers and original studies in the field reported in the last twenty years. The proof of that, is that recently a review manuscript focused on this subject was published – DOI: 10.1039/D1MA00456E (Review Article) Mater. Adv., 2021, 2, 7237-7276. Therefore, the manuscript now submitted will not approach other manuscripts that include sol-gel materials doped with other compounds for pH monitoring, since the authors consider that this approach will turn the manuscript extremely confuse.
Critical analysis of results is missing – comparison with similar functional materials? There are many papers on optical pH sol-gel sensors for concrete published but authors are not citing them.
Author’s answer: The authors appreciate the reviewer comments. However, no studies were conducted in concrete. Nevertheless, new information was introduced. All changes are highlighted in yellow.
Reviewer 2 Report
The MS reports the preparation and characterization of two interesting Organic-Inorganic Hybrids (OIH) produced from sol-gel reactions of a polyether amine (Jeffamine THF 170) and 3 glycidoxy-propyltrimethoxysilane (GOTMS), doped with two purposely synthesized, and structurally similar, imidazole compounds. The methodology used is appropriate, even though not rigorously followed. The main concern is that the authors have not addressed the likely reactions of imidazole NH groups with the epoxy groups in GOTMS (see, for instance, Polymer J. vol 37 pp 833-840). These reactions could ne the main cause for the observed increase in electrical resistance, which can be associated with an increase in cross-linking density. At the same time authors have not considered the possibility of the formation of silsequioxane gages within the networks, which would reduce the number of SiOH groups present (see, for instance, Polymer 2015 vol 63 pp 222-229), which would also increase the electrical resistance. Nevertheless, some useful data have been generated from the work which should not go to waste. A careful re-examination of the results and a substantial revision of the presentation of the work could ultimately produce a valuable addition to the literature.
The comments on specific aspects of the presented work that could be used for a Major revision are outlined below:
- Introduction - Clarify the novelty of the work without quoting results and indicate the expected difference and/or advantages of work already done in this area.
- Materials - a) Show or describe the structure of Jeffamine THF 170. b)Indicate that the molar ratio 1:3 for the Jeff : GOTMS combination corresponds to a NH2: Epoxy ratio = 1:1.5 and therefore at full conversion one expects a total consumption of epoxy groups. c) Clarify the meaning of the statement in Lines 161-162 “….. for 15 days to ensure curing of the film….”
- Results and Discussion - a)The statement in Lines 227-228 is a meaningless generalization “…..incorporation of organic molecules within this type of structure….” and probably in discordance with the actual results. A similar observation applies to statement in Line 234. b) Clarify that the imidazole compounds were introduced before gelation and state the molar ratios with respect to the GOTMS component, corresponding to the addition of different quantities of the solution, 500mL, 600mL, etc. c) Introduce in both FTIR data, Figures 3 and 4, the trace for the control i.e. pure OIH, and identify (possibly) the more usual absorption band for the epoxy in GOTMS at 916 cm-1 (see Polymer 2015 vol; 37 pp 222-229), It should be possible to estimate the degree of conversion of epoxy groups by normalizing this band with respect with an internal invariant, such as the absorption for the C-Si bond at 1245-1265 cm-1. d) Add the UV/Vis spectrum (Fig 5) for the pure OIH system (control) and discuss in relation to the corresponding doped 3a and 3b systems. d) the data for the control (pure) OIH have to be introduced also in the remaining Figures 6-9 and related tables.
- Conclusions - These need to be re-written completely. At present these consist, at the best, of vague statements about what was done and what was obtained and an added-on statement that have a “ … potential to be applied for pH between 9 and 12.5 which are attractive properties to be used as sensor membranes 388 in concrete structures”. Please indicate which data verifies this conclusion.
Author Response
The authors would like to acknowledge all the reviewers involved for their time and dedication in reviewing the article entitled “Hybrid sol-gel matrices doped with colorimetric/fluorimetric imidazole derivatives”. All the comments are very valuable and were considered carefully in order to improve the quality and readability of the manuscript.
Reviewer 2 comments:
The manuscript “Hybrid sol-gel matrices doped with colorimetric/fluorimetric imidazole derivatives” by Figueira et al. presents the synthesis of imidazole derivatives and its immobilization in an hybrid material. Their optical properties were studied by UV-Vis and fluorescence spectroscopies, showing potential application as pH sensor. After the OIH preparation their electronic properties were also evaluated. The manuscript is interesting since presents new insights concerning photoactive OIHs. The spectroscopic characterization is well done and concerning the Nanomaterials policy I should indicate this manuscript for publication. Nevertheless, I find that the impact of the work could considerably be augmented by some small changes in the manuscript, and questions that must be answered by the authors, as follows:
The MS reports the preparation and characterization of two interesting Organic-Inorganic Hybrids (OIH) produced from sol-gel reactions of a polyether amine (Jeffamine THF 170) and 3 glycidoxy-propyltrimethoxysilane (GOTMS), doped with two purposely synthesized, and structurally similar, imidazole compounds. The methodology used is appropriate, even though not rigorously followed. The main concern is that the authors have not addressed the likely reactions of imidazole NH groups with the epoxy groups in GOTMS (see, for instance, Polymer J. vol 37 pp 833-840). These reactions could ne the main cause for the observed increase in electrical resistance, which can be associated with an increase in cross-linking density.
Author’s answer: The possible reaction of imidazole NH groups with residual epoxy groups in GOTMS was now addressed and pointed as a possible cause of electrical resistance increase. All the changes are highlighted in yellow.
At the same time authors have not considered the possibility of the formation of silsequioxane gages within the networks, which would reduce the number of SiOH groups present (see, for instance, Polymer 2015 vol 63 pp 222-229), which would also increase the electrical resistance.
Author’s answer: The possible formation of silsequioxane cages was now addressed and pointed as a possible cause of electrical resistance increase. All changes are highlighted in yellow.
Nevertheless, some useful data have been generated from the work which should not go to waste. A careful re-examination of the results and a substantial revision of the presentation of the work could ultimately produce a valuable addition to the literature.
The comments on specific aspects of the presented work that could be used for a Major revision are outlined below:
- Introduction - Clarify the novelty of the work without quoting results and indicate the expected difference and/or advantages of work already done in this area.
Author’s answer: The authors agree with the reviewer. The last paragraph of the introduction of the manuscript was rewritten focus on the expected difference and/or advantages of work already done in this area. All the changes are highlighted yellow through the manuscript.
- Materials - a) Show or describe the structure of Jeffamine THF 170. b)Indicate that the molar ratio 1:3 for the Jeff : GOTMS combination corresponds to a NH2: Epoxy ratio = 1:1.5 and therefore at full conversion one expects a total consumption of epoxy groups. c) Clarify the meaning of the statement in Lines 161-162 “….. for 15 days to ensure curing of the film….”
Author’s answer: a) The structure of Jeffamine THF 170 was described in Section 3.2. b) The molar ratio for the Jeff:GOTMS is 1:2, corresponding to a NH2:Epoxy ratio of 1:1. This was clarified in Section 3.2, as well as the expected total consumption of epoxy groups. C) The statement was misleading, and it was changed. All the changes are highlighted yellow through the manuscript.
- Results and Discussion - a)The statement in Lines 227-228 is a meaningless generalization “…..incorporation of organic molecules within this type of structure….” and probably in discordance with the actual results. A similar observation applies to statement in Line 234.
Author’s answer: The authors agree with the reviewer. The sentence was misleading and it was changed. All the changes are highlighted yellow through the manuscript.
- b) Clarify that the imidazole compounds were introduced before gelation and state the molar ratios with respect to the GOTMS component, corresponding to the addition of different quantities of the solution, 500mL, 600mL, etc.
Author’s answer: The authors appreciate the reviewer comments. However, the reaction between precursors is performed before the addition of the imidazole compounds. Even though there could be some epoxy groups that do not react with the organic precursor, there is no way of quantifying this ratio since stoichiometric molar ratio of 1 Jeffamine:2 GPTMS was used.
- c) Introduce in both FTIR data, Figures 3 and 4, the trace for the control i.e. pure OIH, and identify (possibly) the more usual absorption band for the epoxy in GOTMS at 916 cm-1 (see Polymer 2015 vol; 37 pp 222-229), It should be possible to estimate the degree of conversion of epoxy groups by normalizing this band with respect with an internal invariant, such as the absorption for the C-Si bond at 1245-1265 cm-1.
Author’s answer: The authors introduced the trace for Jeffamine THF 170 and for the pure film (A(170)). All changes are highlighted in yellow.
- d) Add the UV/Vis spectrum (Fig 5) for the pure OIH system (control) and discuss in relation to the corresponding doped 3a and 3b systems.
Author’s answer: The authors appreciate the reviewer comments. However, the pure matrix was introduced. All changes are highlighted in yellow.
- e) the data for the control (pure) OIH have to be introduced also in the remaining Figures 6-9 and related tables.
Author’s answer: The authors appreciate the reviewer comments. A new figure was introduced – Figure 5 with the Nyquist plot obtained for the pure OIH pure matrices. Additionally, in tables 1 and 2 the information regarding the pure OIH were introduced.
Conclusions – These need to be re-written completely. At present these consist, at the best, of vague statements about what was done and what was obtained and an added-on statement that have a “ … potential to be applied for pH between 9 and 12.5 which are attractive properties to be used as sensor membranes 388 in concrete structures”. Please indicate which data verifies this conclusion.
Author’s answer: The authors agree with the reviewer. The conclusion was rewritten and the mentioned statement was removed. All changes are highlighted in yellow.
Reviewer 3 Report
The manuscript “Hybrid sol-gel matrices doped with colorimetric/fluorimetric imidazole derivatives” by Figueira et al. presents the synthesis of imidazole derivatives and its immobilization in an hybrid material. Their optical properties were studied by UV-Vis and fluorescence spectroscopies, showing potential application as pH sensor. After the OIH preparation their electronic properties were also evaluated. The manuscript is interesting since presents new insights concerning photoactive OIHs. The spectroscopic characterization is well done and concerning the Nanomaterials policy I should indicate this manuscript for publication. Nevertheless, I find that the impact of the work could considerably be augmented by some small changes in the manuscript, and questions that must be answered by the authors, as follows:
- I suggest to remove the sentences from the abstract (Organic-inorganic hybrids ... for fuel cells or cation/pollutants adsorption tumors). The removed sentences must be presented in the introduction section. Please highlight in the abstracts the main results of this investigation. In addition, the statement "following the group’s works with imidazole derivatives for optical applications" must also be removed.
- After carefully read the experimental section, I am skeptical about the affirmation that the obtained material "were immobilized in a well-known OIH sol-gel matrix". Why the authors affirm that the compounds are immobilized into the silica network if their addition was performed after the gelification? (as I could understand from Figure 2). Why these compounds were not added at the beginning to afford a better dispersion in the sol-gel? Finally, Dopped should be better than immobilized.
- Please present the 13C data with only one significant digit after comma.
- To broaden background about imidazole applications please see DOI: 10.1002/tcr.202100138.
-What is the nature of the observed electronic transitions (Figure 1)?
- Absorbance is adimensional, so (a.u.) is nonsense. In addition, Figure 7 is not normalized as presented in the Y-axis. Please change "zone" by "region".
- The authors must discuss why the fluorescence emission of the OIH were taken at different WL?
- In the conclusions the authors affirm that "Further studies must be conducted to test the performance of the material in contact with pH changes ". However, in the abstract is presented that these materials are potential for pH sensing. How did the authors can affirm the last sentence without any proof-of-concept regarding these materials and pH?
- It is not clear the additional applications (sensor membranes in concrete structures) claimed by the authors in the abstract. The text must be improved to clarify this topic. I could not find any clear element to support this affirmation.
Author Response
The authors would like to acknowledge all the reviewers involved for their time and dedication in reviewing the article entitled “Hybrid sol-gel matrices doped with colorimetric/fluorimetric imidazole derivatives”. All the comments are very valuable and were considered carefully in order to improve the quality and readability of the manuscript.
Reviewer 3 comments
- I suggest to remove the sentences from the abstract (Organic-inorganic hybrids ... for fuel cells or cation/pollutants adsorption tumors). The removed sentences must be presented in the introduction section. Please highlight in the abstracts the main results of this investigation. In addition, the statement "following the group’s works with imidazole derivatives for optical applications" must also be removed.
Author’s answer: The authors appreciate the reviewer comments. The sentences were removed, and the main results were introduced as suggested by the reviewer. All the changes are highlighted in yellow.
- After carefully read the experimental section, I am skeptical about the affirmation that the obtained material "were immobilized in a well-known OIH sol-gel matrix". Why the authors affirm that the compounds are immobilized into the silica network if their addition was performed after the gelification? (as I could understand from Figure 2).
Author’s answer: The authors appreciate the reviewer comment. The OIH sol-gel matrices were doped with the compounds, during the gelification. This was corrected through the manuscript.
- Why these compounds were not added at the beginning to afford a better dispersion in the sol-gel? Finally, Dopped should be better than immobilized.
Author’s answer: The authors appreciate the reviewer comment. The compounds were added during the gelification, since this process can occur during 15 days. The dopants were added at this point to afford the best possible dispersion in the matrix, having in mind that the addition could not be performed earlier to avoid the reaction of imidazole group with the epoxy group of 3-GPTMS. This was clarified in the manuscript and all changes are highlighted in yellow.
- Please present the 13C data with only one significant digit after comma.
Author’s answer: The 13C data was changed. All the changes are highlighted in yellow.
- To broaden background about imidazole applications please see DOI: 10.1002/tcr.202100138.
Author’s answer: The authors appreciate the reviewer comments. The article suggested was cited in the introduction since it is a very good addition to the manuscript.
-What is the nature of the observed electronic transitions (Figure 1)?
Author’s answer: Upon interaction with alkaline environment, the bands that are disturbed are those related with the electronic transitions of the non-bonding electrons at the nitrogen atoms, namely the n->π* transitions at longer wavelengths. In Fig 1, the pH-induced deprotonation leads to higher electron density at the imidazole ring and the corresponding bands are shifted to longer wavelengths. This information was introduced and is highlighted in yellow.
- Absorbance is adimensional, so (a.u.) is nonsense. In addition, Figure 7 is not normalized as presented in the Y-axis. Please change "zone" by "region".
Author’s answer: The authors agree with the reviewer and the suggested changes were performed through the manuscript. All changes are highlighted in yellow.
- The authors must discuss why the fluorescence emission of the OIH were taken at different WL?
Author’s answer: The discussion was introduced in the manuscript. All changes are highlighted in yellow.
- In the conclusions the authors affirm that "Further studies must be conducted to test the performance of the material in contact with pH changes ". However, in the abstract is presented that these materials are potential for pH sensing. How did the authors can affirm the last sentence without any proof-of-concept regarding these materials and pH?
Author’s answer: The authors appreciate the reviewer comments and removed the mentioned statement. All changes are highlighted in yellow.
- It is not clear the additional applications (sensor membranes in concrete structures) claimed by the authors in the abstract. The text must be improved to clarify this topic. I could not find any clear element to support this affirmation.
Author’s answer: The authors appreciate the reviewer comments and removed the mentioned statement. All changes are highlighted in yellow.
Round 2
Reviewer 1 Report
Revised manuscript has not addressed the main points raised in my first report, in particular the ones related to pH characterisation methodologies.
A functional material should demonstrate functionality. By doping a sol-gel material with a pH sensitive colorimetric or fluorimetric indicator, one would expect at the absolute minimum that the authors characterise the pH sensitivity of the material and find out whether organic molecules retained their pH response upon immobilisation. Why is this simple experiment missing from this and other similar publications from the group?
In the context of this manuscript it is very important to stress whether doped materials are responsive to pH changes or not. Without demonstrating pH sensitivity of sol-gel membranes (as I already stressed in the first report) this paper should not be published and the authors certainly cannot claim again, in their revised version:
Overall, the results reported herein set the roots for the development of optical fiber sensors functionalized with OIH sol-gel membranes doped with imidazole derivatives for pH monitoring in concrete structures.
Author Response
The authors appreciate the reviewer’s comments. However, pH sensitivity of the films was not addressed since the paper focus on synthesis and characterization of new materials only and that is clearly patent in the manuscript.
Reviewer 2 Report
Authors have addressed most of the issues that I have raised. However it is still not clear why you should get a higher cross linking density if the imidazole is added after the reaction of the epoxy is complete. My comment was that one can only add the imidazole before gelation (a gel does not flow and will not allow dispersion of the imidazole by mixing). Moreover, the suggested increase in cross linking density can only occur through an acceleration of the reactions leading to the formation of the network.
Author Response
The authors appreciate the reviewer’s comments. In fact, imidazole is added after the first reaction between precursors Jeffamine THF-170 and 3-GPTMS. However, 3-GPTMS may have not achieved full consumption and some epoxy groups may have reacted with the imidazole to increase the cross linking density. Since the amount of imidazole added is very small when compared to both precursors, a small percentage of epoxy groups that do not react with Jeffamine would still outnumber the imidazole added. The dopant is added during the sol-gel reaction, so the mixture is not yet a gel, allowing to a dispersion of the imidazole. To form a gel it take at least 2h hours and the adition is performed after 30 minutes (section 2.3).
Reviewer 3 Report
One last question, it is possible to calculate the epsilon from the UV-Vis curves? Based on the shape, position, and mainly intensity (absorbance), I believe that the authors are not dealing at all with n to pi star transitions. It seems that the electronic transition is related to pi to pi star. Please note that the literature already indicated that in these compounds, the HOMO to LUMO transitions is related to the double bonds. Please revise.
Author Response
The authors appreciate the reviewer’s valuable comments. Indeed, ε values were calculated and found to be in the same magnitude order (3b spectra: ε309 = 16586; ε413 = 22439 / 3b+SCPS spectra: ε330 = 16944; ε482 = 13919). This may suggest that the nature of the electronic transition is the same, and the fact that all ε values are larger than 10000 suggest that π-> π* is the electronic transition in both cases. The text in the manuscript was changed and all changes are highlighted in yellow.